# Helios Expression in Tumor-Infiltrating Lymphocytes Correlates with Overall Survival of Advanced Gastric Cancer Patients

**DOI:** 10.3390/life10090189

**Published:** 2020-09-10

**Authors:** Wei-Ming Chen, Jing-Lan Liu, Huei-Chieh Chuang, Yong-Lin Chang, Chia-Ming Yeh, Cheng-Shyong Wu, Shu-Fen Wu

**Affiliations:** 1Division of Gastroenterology and Hepatology, Department of Internal Medicine, Chang Gung Memorial Hospital, Chiayi 613016, Taiwan; b8902030@gmail.com; 2College of Medicine, Chang Gung University, Taoyuan 333323, Taiwan; 3Graduate Institute of Clinical Medical Sciences, College of Medicine, Chang Gung University, Taoyuan 333323, Taiwan; 4Department of Pathology, Chang Gung Memorial Hospital, Chiayi 613016, Taiwan; fen221@cgmh.org.tw (J.-L.L.); a9287@cgmh.org.tw (H.-C.C.); 5Department of Biomedical Sciences, and Institute of Molecular Biology, National Chung-Cheng University, Min-Hsiung, Chiayi 621301, Taiwan; shulie168@yahoo.com.tw (Y.-L.C.); yeh-chiaming@hotmail.com (C.-M.Y.); 6Center for Innovative Research on Aging Society (CIRAS), National Chung Cheng University, Chiayi 621301, Taiwan

**Keywords:** gastric cancer, tumor-infiltrating lymphocyte, Helios, IKZF2

## Abstract

Immunotherapy is a highly promising approach for the treatment of gastric cancer, the third-leading cause of overall cancer death worldwide. In particular, tumor-infiltrating lymphocytes and peripheral blood mononuclear cells are believed to mediate host immune responses, although this activity may vary depending on the activation status and/ or their microenvironments. Here, we examined the expression of a specific zinc finger transcription factor, Helios (IKZF2), in gastric tumor-infiltrating lymphocytes by immunohistochemistry and the correlation with survival. Segregation of gastric cancer patients into high- vs. low-Helios-expressing tumor-infiltrating lymphocytes showed those with high expression to exhibit longer survival in gastric cancer patients, Helicobacter pylori-infected gastric cancer patients and advanced stage (III–IV) gastric cancer patients. In particular, Helios expression was an independent factor for survival in advanced gastric cancer patients. We performed immunofluorescence staining to detect Helios expression in tumor-infiltrating lymphocytes and peripheral blood mononuclear cells. We found that Helios is expressed more in CD4+ T cells and little in CD8+ T cells in infiltrated lymphocytes in gastric cancer. In summary, we believe that the study of specific characteristics of tumor-infiltrating lymphocytes can delineate the interactions of immune and tumor cells to improve upon immunotherapy strategies.

## 1. Introduction

While gastric cancer (GC) is the third-leading cause of global cancer-related deaths in 2018, it varies widely by geography, with the highest incidence rate in East Asia [1]. At the tissue level, the GC tumor microenvironment is also a predominant factor that impacts tumor development and therapeutic effects [2,3], and includes both cancerous and non-cancerous cells (e.g., stromal cells and immune cells) [4]. Both innate and adaptive immune cells normally present in the gastrointestinal tract have been reported to play important roles in the GC tumor microenvironment, either inhibiting or enhancing tumor development. Recently, cancer immunotherapy has brought a breakthrough to cancer treatment, yielding remarkable clinical outcomes through reawakening host antitumor immunity [5]. Tumor-infiltrating lymphocytes (TILs) are crucial factors of the tumor microenvironment and reflect host antitumor immune responses. It has been reported that high levels of TILs reflected a protective role of host in antitumor immunity against GC and was associated with positive prognosis [6]. Additionally, CD8+ and CD4+ T cells, two subpopulations of CD3+ T cells, play different roles in tumor immune responses [7]. The activated CD8+ T cells become cytotoxic T cells, which play a crucial role in determining the clinical outcomes of tumor patients. CD4+ T cells secretes cytokines such as interferon-γ to activate CD8+ T cells and macrophages. Regulatory T (Treg) cells are one particular subset of CD4+ T cells that are essential for immune suppressive function by inhibiting other effector T cell functions. All these cells interact with others and shape the GC tumor microenvironment.

Ikaros family members belong to zinc finger transcription factors and play important roles in cell fate decisions in hematopoiesis, and especially in lymphocyte development [8,9]. These have also been shown to act as tumor suppressors, because their misexpression in leukemia associates with poor prognosis [10]. One such Ikaros family member, Helios, is expressed in T cells, and in particular, a subpopulation of Foxp3+ Treg cells [11]. Enforced expression of Helios can enhance immunosuppressive function of Foxp3+ CD4+ T cells [12], while Helios-deficient Treg cells fail to control the expansion of pathogenic T cells derived from scurfy mice [13]. Helios also regulates chromatin accessibility during self-renewal in leukemic stem cells and inhibits myeloid differentiation gene expression [14]. Furthermore, Helios interacts with nucleosome remodeling and histone deacetylase complexes, indicating that Helios can regulate transcription by epigenetic mechanisms [15]. Besides Treg cells, Helios is also expressed in a small subset of CD4+ Foxp3− and CD8+ T cells, and Helios expression is induced during T cell activation and proliferation [16,17]. In our previous study, we also found that Helios expressed both in Foxp3+ and Foxp3− lymphocytes infiltrated in gastric cancer [18]. In the current study, we examined Helios expression in gastric tumor-infiltrating lymphocytes and the correlation with prognosis in gastric cancer patients.

## 2. Materials and Methods 

### 2.1. Tissue Samples

Gastric cancer samples from 67 pathologically diagnosed patients (47 males and 20 females; median age, 69 years; range 46–87 years), were collected at Chang Gung Memorial Hospital (CGMH), Chia-Yi, Taiwan. The pathologic examination and further investigations were carried out after obtaining informed consent from patients. All human subject assessments were approved by the Institutional Review Board (IRB) of Chang Gung Memorial Hospital (IRB NO.201700168A3).

### 2.2. Immunohistochemistry (IHC)

Formalin-fixed and paraffin-embedded tissue specimens were cut into 4-μm-thick sections. The samples were heated at 65 °C for 30 min, deparaffinized in xylene, and rehydrated using a 100–75% gradient of ethanol. The slides were then washed three times for 3 min in phosphate-buffered saline (PBS). After antigen retrieval and blocking, the samples were then incubated with a rabbit polyclonal anti-Helios antibody (Sigma-Aldrich^®^ Life Sciences, St. Louis, MO, USA) at 4 °C overnight. The treated slides were then subsequently washed three times for 3 min with PBS, and the samples were detected by a Super SensitiveTM Polymer-horseradish peroxidase (HRP) IHC Detection System (BioGenex, San Ramon, CA, USA), according to the manufacturer’s protocol. Finally, the slides were counterstained with hematoxylin before mounting.

The intensity, percentage, and subcellular localization of immunohistochemical staining for each case were recorded. The intensity of staining was recorded as 0, 1, 2, and 3, referring to negative, weak, moderate, and strong staining, respectively. The percentages of positive cells were recorded from 0% to 100%. The results of staining were scored using a quick (Q) score, which was obtained by multiplying the percentage of positive cells (P) by the intensity (I) (Q  =  P × I; maximum  =  300) [19]. Pathologists evaluated the results of immunohistochemical staining without knowledge of the clinicopathologic data.

For semi-quantifying Helios expression, the optimal cutoff points of the Q scores were determined using X-Tile 3.6.1 software (Yale University, New Haven, CT, USA), as previously described [20]. This program calculates chi-square values at all possible divisions based on the log-rank test for Kaplan–Meier estimates. The cutoff points of the Q scores, with the highest chi-square values, were 160 and 60 for Helios in tumor cells and stromal leukocytes, respectively. For CD4+ lymphocytes in the tumor stroma, 50% was used as a cutoff value. The others were classified as negative.

### 2.3. Double Staining Immunofluoresecence

Immunofluorescence studies of paraffin-embedded tissue specimens were carried out using the same steps used in the IHC procedure, employing an anti-IKZF2 rabbit polyclonal (HPA059142, Sigma-Aldrich, St. Louis, MO, USA), anti-CD3 rabbit monoclonal (ab16669, Abcam, Cambridge, UK), and anti-CD8 mouse monoclonal (ab17147, Abcam, Cambridge, UK) antibodies, at 4 °C overnight. Fluorescence detection antibodies, including goat anti-rabbit Alexa fluor 647 (ab150079, Abcam, Cambridge, UK) and goat anti-mouse Alexa flour 488 (A28175, Thermo Fisher Scientific, Waltham, MA, USA) were incubated with the slides for 1 h at 37 °C. DAPI (564907, BD Biosciences, Franklin Lakes, NJ, USA) was then added for 15 min at room temperature. Subsequent analysis of CD3, CD4, CD8, and Helios distribution in gastric cancer samples was performed by confocal microscopy (Olympus IX81, Olympus Corporation, Tokyo, Japan).

### 2.4. Flow Cytometry

Ficoll–Paque gradient centrifugation isolated peripheral blood mononuclear cells (PBMCs) were washed twice with staining buffer (2% FBS in PBS), followed by staining with fluorescence dye-conjugated specific antibodies (anti-human CD4-APC or CD8-PEcy5) at 4 °C for 30 min in the dark, and they were then washed twice with PBS. Before intracellular staining, cells were incubated with Cytofix/Cytoperm buffer (BD Biosciences) at 4oC for 15 min and were then washed twice with Perm/Wash buffer (BD Biosciences). Hamster IgG-PE or anti-human Helios-PE antibodies were used for intracellular staining. Cell marker was then analyzed by FACSCalibur flow cytometry, using CellQuest (BD Biosciences) and FlowJo (BD Biosciences) software.

### 2.5. Statistical Analysis

The Mann–Whitney test was used to assess the associations with IHC score of Helios. The Kaplan–Meier method was used to construct the overall survival curves, and a log-rank test was used to assess the significance of differences in survival. All statistical analyses were performed using SPSS software version 18.0 (SPSS, Inc., Chicago, IL, USA) or GraphPad Prism 6 (GraphPad Software, Inc., San Diego, CA, USA). *p* < 0.05 was considered statistically significant.

## 3. Results

### 3.1. Higher Expression of Helios in Tumor-Infiltrating Lymphocytes Correlates with Better Overall Survival in Gastric Cancer Patients and Helicobacter Pylori-Positive Gastric Cancer Patients

Tumor-infiltrating lymphocytes are crucial factors of the tumor microenvironment and reflect host antitumor immune responses. To detect Helios expression in TILs, we performed immunohistochemistry using an anti-Helios antibody in GC samples (Figure 1, upper panel: anti-Helios, lower panel: H&E stain). Expression levels of Helios in TILs were defined in 67 GC tumor patients as a high Helios expression group (*n* = 45) and a low expression group (*n* = 22) according to immunohistochemistry. Low Helios expression in gastric cancer are shown in Figure 1A,B (Figure 1A: 100X, 1B: 400X), and the consecutive slide with H&E stain are shown in Figure 1E,F (Figure 1E: 100X, Figure 1F: 400X). High Helios expression in gastric cancer is shown in Figure 1C,D (Figure 1C: 100X, 1D: 400X), and the consecutive slide with H&E stain is shown in Figure 1G,F (Figure 1E: 100X, Figure 1F: 400X). Helios was expressed not only in tumor-infiltrating cells, but also in in epithelial cells. The patients’ characteristics are shown in Table 1 according to their association with Helios expression in TILs. A Kaplan–Meier survival analysis revealed significantly better overall survival in the high Helios expression group than in the low Helios expression group (Figure 2A, log-rank test, *p* = 0.019). Chronic infection with Helicobacter pylori (H. pylori, HP) is an important risk factor, and, notably, the cagA-positive strain can enhance the gastric malignancy [21]. In a subgroup of H. pylori-positive infected GC patients, the high Helios-expressing TIL group showed significantly longer survival than those in the low Helios-expressing TIL group (Figure 2B, log-rank test, *p* = 0.02). However, in HP-negative patients, the high vs. low Helios expression levels did not differentially affect survival (Figure 2C, log-rank test, *p* = 0.238).

### 3.2. Helios Expression in Tumor-Infiltrating Lymphocytes Was an Independent Factor for Survival in Advanced Gastric Cancer Patients

We also analyzed the effect of Helios expression in TILs in early (stage I + II) and advanced (stage III + IV) tumor-node-metastasis (TNM) cancer stages as associated with overall survival. To this end, Kaplan–Meier survival analysis revealed that high Helios expression in TILs is a significantly better prognostic factor of median overall survival than low Helios expression (Figure 3A, log-rank test, *p* = 0.029) in advanced stage GC patients, but not in earlier stage patients (Figure 3B, log-rank test, *p* = 0.977). In the subgroup study of advanced gastric cancer patients, there was no significant difference of patient characteristics in age, gender, LN metastasis, distant metastasis, HP infection, tumor-infiltrated CD4 T cells and tumor-infiltrated Foxp3+ T cells (Table 2). In a multivariate analysis, the hazard ratio for overall survival among the high Helios subjects versus the low Helios subjects was 2.90 in advanced gastric cancer patients (Table 3, 95 % confidence interval (CI), 1.02–8.20; *p* = 0.046). Other factors included gender, distant metastasis, HP infection, tumor infiltrated CD4 T cells and tumor-infiltrated Foxp3+ T cells, which did not show statistical significance in overall survival.

### 3.3. The Expression of Helios, CD3, CD4, and CD8 in Gastric Tumor-Infiltrating Lymphocytes

To detect Helios expression in TILs, we performed immunofluorescence staining of gastric tumor tissues. CD4 and CD8 antibodies were used to distinguish the two major subpopulations of CD3-positive T cells (CD3 and CD4, Figure 4A; CD3 and CD8, Figure 4C). Double staining of Helios and T cells showed that partial CD4 T cells expressed Helios, but very little CD8 T cells did (Figure 4B: Helios and CD4, Figure 4D: Helios and CD8). Among T cells, Helios was partially co-expressed with CD4, but very little with CD8 (Figure 4B,D). Except for immunofluorescence staining of patient samples, PBMCs were examined for expression of CD4 and CD8 in association with Helios (Figure 4E). Both CD4+ and CD8+ T cells were coexpressed with Helios in low percentages, and the expression of Helios in CD4+ T cells was higher than those in CD8+ T cells in PBMCs. 

## 4. Discussion

In this study, we demonstrated that high expression of Helios in tumor-infiltrating lymphocytes had a better median overall survival in gastric cancer patients, Helicobacter pylori-infected patients (Figure 2) and advanced gastric cancer patients (Figure 3). In particular, from multivariate analysis, the Helios expression in tumor-infiltrating lymphocytes was an independent factor to predict better survival in advanced gastric cancers (Table 3). 

Helios belongs to the Ikaros transcription factor family, characterized by a highly conserved C2H2 zinc-finger DNA-binding domain near the N terminus, and a second zinc finger protein–protein interaction domain in the C terminus [9,22]. Ikaros, the founding member of this family, is mainly expressed in most hematopoietic cells, whereas Helios is expressed primarily in T-lineage cells and early multipotential precursor cells [23,24]. Helios is expressed in 60–70% of Treg cells in both mice and humans [25]. However, in mice, specific deletion of Helios in Treg cells developed systemic autoimmunity; in contrast, specific deletion in CD4 cells lacks an autoimmune phenotype [26]. In previous studies, Helios is expressed in a small subset of CD4+ Foxp3− and CD8+ T cells, and the expression is associated with T cell activation and proliferation both in CD4+ and CD8+ T cells [16,17]. In fact, Helios is expressed in a small group of CD4+ Foxp3− T cells with memory phenotype in vivo, and the Helios-deficient naïve CD4 cells differentiate into pTreg cells after specific antigen stimulation to suppress effector cell functions [17]—Helios probably plays a significant functional role in normal CD4+ T-cell activation. In this study, we found that higher Helios expression in advanced gastric cancer patients with better prognosis. We speculated that during the repetitive gastric tumor antigen stimulation, higher expression of Helios represented more T cell activation or more memory T cells, which will provide more active immune responses to against gastric cancer. However, this phenomenon needs further investigation.

During the tumorigenesis, normal cells are transformed into tumor cells, a transformation which is characterized by changes at the cellular, genetic, and epigenetic levels, as well as abnormal cell division. In the tumor microenvironment, chronic inflammation has been indicated with tumor developments, and both effector T cells and Treg cells play important roles to modulate tumor development. In malignancies such as GC, heterogeneity is often observed in tumor cells known to establish immunosuppressive environments. To that effect, tumor-intrinsic genetic and epigenetic alterations induce tumor cells to produce suppressive molecules within tumor microenvironments, facilitating escape from immune attack and representing a major impediment to cancer immunotherapies’ anti-tumor immune responses [27]. Recently, it has been shown that cancer immunotherapy “reawakes” anti-tumor immune responses, and can even cure metastatic disease and yield remarkable clinical outcomes. On the other hand, microsatellite instability (MSI) is the condition of genetic hypermutability caused by impaired DNA mismatch repair and is often found in GC [28]. MSI tumors elicit a strong immunoresponse, and patients with MSI-high GC show a significant longer overall survival compared with those who have microsatellite-stable features [29]. As noted above, the MSI condition in addition to Helios may need to be checked in GC patients.

With regard to GC pathogenesis, Helicobacter pylori is its best-understood risk factor, with gastric biopsy showing increased numbers of Foxp3+ Tregs in H. pylori-infected gastritis and GC patients [30]. For GC, several studies strongly indicate that immune cells contribute to determining prognosis or overall survival [6,31,32]. However, the exact role of immune cells in GC remains unclear. Thus, the immune status of patient tumors is an important parameter for evaluating therapeutic strategies to predict therapeutic response and even forecast long-term survival.

Lymphocytes of the host immune system migrate to the tumor site to attack malignant cells, and lymphocyte subsets can be physically characterized by their surface-antigens, such as cluster of differentiation (CD) markers. Migration of lymphocytes to tumor sites might imply that the host immune system is capable of initiating an antitumor response, and a recent report indicated that anatomical location contributes to the immune constitution of a developing primary tumor [33]. In contrast, the percentage of CD8+ T cells found in cervical carcinomas was similar to that in normal cervical tissues, although the composition of CD4+ T cells did differ. Thus, differences in CD4+ T-cell infiltration are reflected in prognosis of survival [33]. 

During immune responses, CD4+ naive T cells differentiate into distinct lineages that express diverse transcription factors, and they have different developmental pathways and unique biological functions. For instance, CD4+ T naive cells differentiate into T helper 1 (Th1) cells characterized by interferon gamma secretion, which can help CD8+ T cells convert into cytotoxic T cells. However, deficiency of of Helios increases pTreg differentiation from CD4+ T naive cells, causing a reduction of effector T cells. Moreover, many suppressor cells such as myeloid-derived suppressor cells, tumor associated macrophages or neutrophils all can play some roles in the constitution of the tumor microenvironment to inhibit effective immune responses. In summary, studies of factors that mediate immune system action, such as Helios, will provide better insight into immunotherapy for cancer and other pathologies. 

## Figures and Tables

**Figure 1 life-10-00189-f001:**
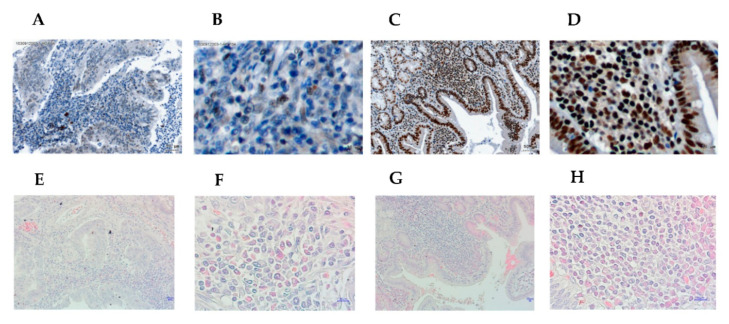
Immunohistochemical and H&E stains in gastric tumor. (**A**–**D**) Immunohistochemical staining using the anti-Helios antibody in gastric tumor tissues. Low expression of Helios in gastric cancer is represented by (**A**) (100X) and (**B**) (400X). High expression of Helios in gastric cancer is represented by (**C**) (100X) and (**D**) (400X). (**E**–**H**) H&E staining in gastric tumor tissues. Low expression of Helios in gastric cancer is represented by (**E**) (100X) and (**F**) (400X). High expression of Helios in gastric cancer is represented by (**G**) (100X) and (**H**) (400X). The scale bar in (**A**) and (**C**) is 50 μM, and others it is 10 μM.

**Figure 2 life-10-00189-f002:**
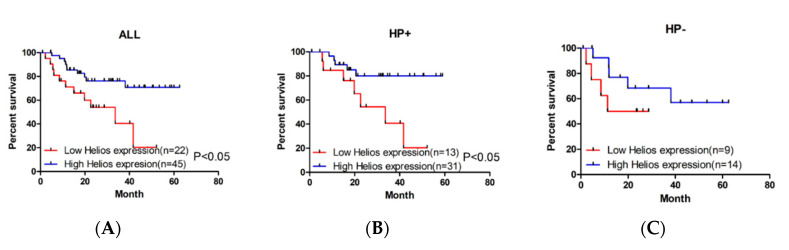
Kaplan–Meier curves and log-rank test results for the overall survival of the patients with gastric cancer based on Helios expression in tumor-infiltrating lymphocytes in tumor tissues. (**A**) Kaplan–Meier curves of overall survival of gastric cancer patients obtained by the expression of Helios in tumor-infiltrating lymphocytes in tumor tissues. The test resulted in a *p* value < 0.05. (**B**) Kaplan–Meier curves of overall survival in H. pylori infected (+) gastric cancer patients obtained by the expression of Helios in tumor-infiltrating lymphocytes in tumor tissues. The test resulted in a *p* value < 0.05. (**C**) Kaplan–Meier curves of overall survival in H. pylori infected (-) gastric cancer patients obtained by the expression of Helios in tumor-infiltrating lymphocytes in tumor tissues.

**Figure 3 life-10-00189-f003:**
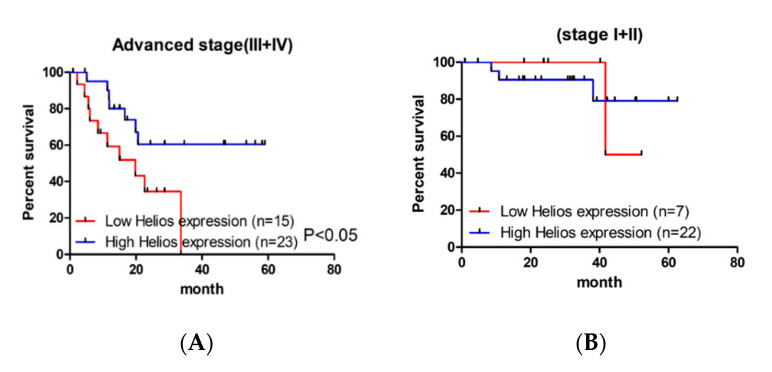
Kaplan–Meier curves and log-rank test results for the overall survival of the patients with advanced gastric cancer based on Helios expression in tumor-infiltrating lymphocytes in tumor tissues. (**A**) Kaplan–Meier curves of overall survival of advanced-stage (T3 + T4) gastric cancer patients obtained by the expression of Helios in tumor-infiltrating lymphocytes in tumor tissues. The test resulted in a *p* value < 0.05. (**B**) Kaplan–Meier curves of overall survival of lower stage (T1 + T2) gastric cancer patients obtained by the expression of Helios in tumor-infiltrating lymphocytes in tumor tissues. No statistic differences were observed.

**Figure 4 life-10-00189-f004:**
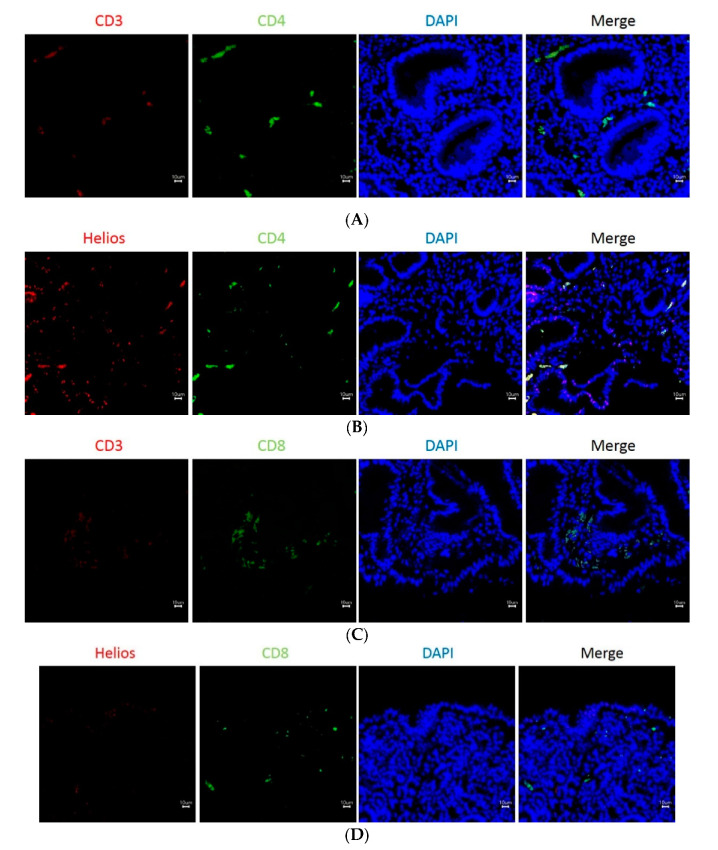
Immunofluorescence staining of infiltrating lymphocytes in gastric cancer. Immunofluorescence staining of the infiltrated lymphocytes was stained with anti-CD3 (**A**,**C**, “CD3”), anti-CD4 (**A**,**B**, “CD4”), anti-Helios (**B**,**D**, “Helios”) and anti-CD8 (**C** and **D**, “CD8”) antibodies. DAPI stains with nucleic acid. (**E**) Expression of CD4+, CD8+, and Helios on PBMCs. Dot plots of fluorescence-activated cell sorting (FACS) analysis on CD4+ cells (left panel) or CD8+ cells (right panel) stained with isotype antibodies (upper panel) or anti-Helios antibodies (lower panel). The scale bar in (**A**–**D**) (100X) is 10 μM.

**Table 1 life-10-00189-t001:** Characteristics of total gastric cancer patients based on Helios presentation.

	All (*n*=67)	High (*n* = 45)	Low (*n* = 22)	*p*
**Age**	68.5 ± 10.0	69.2 ± 10.1	67.1 ± 10.0	0.413
**Gender**				0.415
Male	47	33	14	
Female	20	12	8	
**TNM stage**				0.185
I + II	29	22	7	
Advanced (III + IV)	38	23	15	
**LN metastasis**				0.395
Yes	44	28	16	
No	23	17	6	
**Distal metastasis**				0.047
Yes	10	4	6	
No	57	41	16	
**HP infection**				0.428
Yes	44	31	13	
No	23	14	9	
**CD4**				0.911
High	25	17	8	
Low	42	28	14	
**Foxp3**				0.926
High	31	21	10	
Low	36	24	12	

Abbreviation: TNM: tumor-node-metastasis; LN: lymph node; HP: Helicobacter pylori

**Table 2 life-10-00189-t002:** Characteristics of advanced gastric cancer patients based on Helios presentation.

	All (*n* = 38)	High (*n* = 23)	Low (*n* = 15)	*p*
**Age**	67.7 ± 11.1	68.7 ± 11.5	66.2 ± 10.8	0.507
**Gender**				0.544
Male	25	16	9	
Female	13	7	6	
**LN metastasis**				0.210
Yes	37	23	14	
No	1	0	1	
**Distal metastasis**				0.122
Yes	10	4	6	
No	28	19	9	
**HP infection**				0.311
Yes	24	16	8	
No	14	7	7	
**CD4**				0.851
High	12	7	5	
Low	26	16	10	
**Foxp3**				0.376
High	16	11	5	
Low	22	12	10	

**Table 3 life-10-00189-t003:** Association between hazard rate and risk factors based on the Cox model in advanced gastric cancer patients.

Risk Factors	Hazard Ratio (95% CI)	*p*
**Helios**		0.046
High	1	
Low	2.90 (1.02–8.20)	
**Gender**		0.063
Male	1	
Female	3.50 (0.94–12.99)	
**Distal metastasis**		0.091
Yes	1	
No	3.72 (0.81–16.95)	
**HP infection**		0.239
Yes	1	
No	1.92 (0.65–5.68)	
**CD4**		0.495
High	1	
Low	0.63 (0.16–2.39)	
**Foxp3**		0.901
High	1	
Low	0.93 (0.30–2.87)

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
