# Peer review of "Helios Expression in Tumor-Infiltrating Lymphocytes Correlates with Overall Survival of Advanced Gastric Cancer Patients"

_life, 2020, doi:10.3390/life10090189_

Round 1
Reviewer 1 Report
In this manuscript, the authors examined Helios expression in gastric tumor-infiltrating lymphocytes. They found that Helios is expressed more in CD4+ T cells and little in CD8+T cells in infiltrated lymphocytes in gastric cancer. They suggested that the study of specific characteristics of tumor-infiltrating lymphocytes can delineate the interactions of immune and tumor cells to improve upon immunotherapy strategies. The manuscript is well written and the results are well presented. On the overall, the topic of the current study is interesting.
Minor comments
- The images of Figure 1 and Figure 4 should include scale bar.
- They suggested that the study of specific characteristics of tumor-infiltrating lymphocytes can delineate the interactions of immune and tumor cells to improve upon immunotherapy strategies. Do the authors have any preliminary data or references to support their hypothesis?
Author Response
Minor comments
- The images of Figure 1 and Figure 4 should include scale bar.
Response: We thank the reviewer’s comment, the scale bar has been added in Figure 1 and Figure 4. Figure 1A and 1C showed 50μM scale bar, while others showed 10μM scale bar (on line 188 and line 264).
- They suggested that the study of specific characteristics of tumor-infiltrating lymphocytes can delineate the interactions of immune and tumor cells to improve upon immunotherapy strategies. Do the authors have any preliminary data or references to support their hypothesis?
Response: The compositions of immune cells in tumor play a major role for the clinical outcome of patients. It has been reported that higher level of TILs reflected a protective role of host in antitumor immunity against GC and was associated with positive prognosis1, and also in many other cancers such as colorectal cancer2. Furthermore, TIL can be an independent prognostic factor for disease free survival in the triple negative breast cancer3. Tumor cells use various strategies to evade immune surveillance such as recruitment of immunosuppressive cells (e.g. regulatory T cells, type 2 macrophages and Myeloid-derived suppressor cells) to suppress effective immune responses, the composition of cells in the tumor microenvironment play pivotal roles in immune therapy4. If we can more understand the tumor and immune interaction, we will more improve the immune therapy of cancer patients.
Reference
- Zhang D, He W, Wu C, Tan Y, He Y, Xu B, et al. Scoring System for Tumor-Infiltrating Lymphocytes and Its Prognostic Value for Gastric Cancer. Front Immunol 2019;10:71.
- Galon J, Costes A, Sanchez-Cabo F, Kirilovsky A, Mlecnik B, Lagorce-Pages C, et al. Type, density, and location of immune cells within human colorectal tumors predict clinical outcome. Science 2006;313:1960-1964.
- Schirosi L, Saponaro C, Giotta F, Popescu O, Pastena MI, Scarpi E, et al. Tumor Infiltrating Lymphocytes and NHERF1 Impact on Prognosis of Breast Cancer Patients. Transl Oncol 2020;13:186-192.
- Pitt JM, Marabelle A, Eggermont A, Soria JC, Kroemer G, Zitvogel L. Targeting the tumor microenvironment: removing obstruction to anticancer immune responses and immunotherapy. Ann Oncol 2016;27:1482-1492.

Reviewer 2 Report
Comments to the authors
Helios Expression in tumor-infiltrating lymphocytes correlates with overall survival of advanced gastric cancer patients, Life, 882769
The manuscript from Chen et al. focuses on a scientifically important field. Due to the high mortality rate of solid malignancies in adults the manuscript fits in the scope of Life journal.
The higher expression of Helios was correlated with longer survival in gastric cancer patients. Statistics on human patient-derived specimens is done properly. However, the study relies only on data from IHC, IF, and FACS investigations. The study lacks mechanistic analysis. Because of the lack of mechanistic view, this clinical descriptive study may hardly fulfill the requirements for publishing in Life Journal in the present form.
Major comments:
- Some sentences need thorough grammatic and stylistic revision.
- Helios is known about higher expression in Tregs. These Tregs suppress antigen specific immune responses. In contrast, higher expression of Helios in TILs prolonged survival. This sounds a paradox. The authors should reinforce their conclusion with mechanistic experiments or demonstrate supporting literature data.
- The immunophenotyping of TILs is missing, what kind of T- (within CD4 or CD8 population) or B-cell subsets express high Helios?
- Cite and explain Fig 1A, B, E, F, G, and H in the text.
- There are few significant data, only p=0.047 in Table 1 and p=0.046 in Table 3.
- Figure 4B should be cited and explained in the text.
- Dot plots in Figure 4E looks like background. The authors should use proper controls and demonstrate the staining with an isotype control antibody or use only secondary antibody stained controls (in case of indirect IF). At least the autofluorescence of unstained samples should be demonstrated. The corresponding Methods section lacks data about the antibodies, fluorophores. Which cells were analyzed? How were these cells isolated?
- This discussion poorly strengthens the results of the study. Lines 250-255 are basic textbook data.
Minor comments:
- Line 27: write “worldwide” at the end of the sentence
- Lines 33-38: Divide this content into separate sentences
- Line 38: Delete or rephrase: “Except immunohistochemistry”
- Line 59: “It reported” should be ‘it was reported’ or ‘it has been reported’
- Lines 59-69: Reword that sentence
- Lines 61-64: This is a textbook information, basic knowledge on the field.
- Lines 64-67: Divide this content into separate sentences
- Myeloid cells, such as MDSCs and TAMs are not mentioned. These cells should be mentioned because these have fundamental effect to maintain the tumor promoting microenvironment (e.g. Inflammation and Cancer: Extra- and Intracellular Determinants of Tumor-Associated Macrophages as Tumor Promoters, Mediators of Inflammation 2017. )
- Line 71: Change “They” to These, and “perform” to act
- Change “foxp3” to Foxp3
- Lines 98-100: This sentence is already in the introduction. This does not fit in the Results section.
- Line 105-106, this sentences is not complete, reword it. ……”malignancies especially a cagA-positove strain”….
- Reword the Figure Legend 1.
- Line 162: Check that paradox: „It did not 162 show significant differences. The test resulted in a P value <0.05”. The p value under 0.05 generally means one star significance.
- Line 204: Write human instead of “man”.
Author Response
Major comments:
- Some sentences need thorough grammatic and stylistic revision.
Response: We thank the reviewer’s comment and have revised the manuscript as the reviewer’s suggestions.
- Helios is known about higher expression in Tregs. These Tregs suppress antigen specific immune responses. In contrast, higher expression of Helios in TILs prolonged survival. This sounds a paradox. The authors should reinforce their conclusion with mechanistic experiments or demonstrate supporting literature data.
Response: We thanks the reviewer’s comment and have revised the sentences (on lines 278-299) in discussion as the following:
“Helios is expressed in 60–70% of Treg cells in both mouse and human [25]. However, in mice, specific deletion of Helios in Treg cells developed systemic autoimmunity, in contrast, specific deletion in CD4 cells lack an autoimmune phenotype [26]. In previous studies, Helios is expressed in a small subset of CD4+Foxp3- and CD8+ T cells, and the expression is associated with T cell activation and proliferation both in CD4+ and CD8+ T cells [16, 17]. In fact, Helios is expressed in a small group of CD4+Foxp3- T cells with memory phenotype in vivo, and the Helios-deficient naïve CD4 cells differentiate into pTreg cells after specific antigen stimulation to suppress effector cell functions [17], probably Helios plays a significant functional role in normal CD4+ T-cell activation. In this study, we found that higher Helios expression in advanced gastric cancer patients with better prognosis. We speculated that during the repetitive gastric tumor antigen stimulation, higher expression of Helios represented more T cell activation or more memory T cells, which will provide more active immune responses to against gastric cancer. However, it needs more investigations to explore this phenomenon. During the tumorigenesis, normal cells are transformed into tumor cells which is characterized by changes at the cellular, genetic, and epigenetic levels and abnormal cell division. In tumor microenvironment, chronic inflammation has been indicated with tumor developments, both effector T cells and Treg cells play important roles to modulate tumor development. In malignancies such as GC, heterogeneity is often observed in tumor cells known to establish immunosuppressive environments.”
The immunophenotyping of TILs is missing, what kind of T- (within CD4 or CD8 population) or B-cell subsets express high Helios?
Response: The figure 4 of this manuscript, immunofluorescense staining of gastric tumor tissues showed that partial CD4 T cells expressed Helios, but very little CD8 T cells did (Figure 4B: Helios and CD4, Figure 4D: Helios and CD8). Both CD4+ and CD8+ T cells were coexpressed with Helios, in low percentages, and the expression of Helios in CD4+ T cells was higher than those in CD8+ T cells in peripheral blood mononuclear cells (Figure 4E).
- Cite and explain Fig 1A, B, E, F, G, and H in the text.
Response: We thanks the reviewer’s comment and have revised the sentences (on lines 162-168) as the following:
“Low Helios expression in gastric cancer were shown in figure 1A and 1B (Figure 1A:100X, 1B: 400X), and the consecutive slide with H&E stain were shown in figure 1E and 1F (Figure 1E:100X, Figure 1F: 400X). High Helios expression in gastric cancer were shown in figure 1C and 1D (Figure 1C:100X, 1D: 400X), and the consecutive slide with H&E stain were shown in figure 1G and 1F (Figure 1E:100X, Figure 1F: 400X). Helios was expressed not only in tumor-infiltrating cells, but also in in epithelial cells.”
- There are few significant data, only p=0.047 in Table 1 and p=0.046 in Table 3.
Response: The data collected from clinical patients contain many factors to influence the results, these two tables show the data from gastric cancer patients, however, the p valve is < 0.05, there are statistic differences.
- Figure 4B should be cited and explained in the text.
Response: We thanks the reviewer’s suggestion and have revised the sentences (on lines 242-248) as the following:
“Double staining of Helios and T cells showed that partial CD4 T cells expressed Helios, but very little CD8 T cells did (Figure 4B: Helios and CD4, Figure 4D: Helios and CD8). Among T cells, Helios was partially co-expressed with CD4, but very little with CD8 (Figure 4B and 4D). Except for immunofluorescence staining of patient samples, peripheral blood mononuclear cells (PBMCs) were examined for expression of CD4 and CD8, in association with Helios (Figure 4E).”
- Dot plots in Figure 4E looks like background. The authors should use proper controls and demonstrate the staining with an isotype control antibody or use only secondary antibody stained controls (in case of indirect IF). At least the autofluorescence of unstained samples should be demonstrated. The corresponding Methods section lacks data about the antibodies, fluorophores. Which cells were analyzed? How were these cells isolated?
Response: We thanks the reviewer’s suggestion and have added the isotype control in Figure 4E. Additionally, the sentences in Material and Methods (on lines 136-144) and in the legend of Figure 4 (on lines 261-264) of our manuscript were revised as following:
“Ficoll-Paque gradient centrifugation isolated PBMCs were washed twice with staining buffer (2% FBS in PBS), and followed stained with fluorescence dye-conjugated specific antibodies (anti-human CD4-APC or CD8-PEcy5) at 4oC for 30 min, in the dark, and subsequently washed twice with PBS. Before intracellular staining, cells were incubated with Cytofix/Cytoperm buffer (BD Biosciences) at 4oC for 15 min, and followed were washed twice with Perm/Wash buffer (BD Biosciences). Hamster IgG-PE or anti-human Helios-PE antibodies were used for intracellular staining. Cell marker was then analyzed by FACSCalibur flow cytometry, using CellQuest (BD Biosciences) and FlowJo (BD Biosciences) software. ”
“Dot plots of fluorescence-activated cell sorting (FACS) analysis on CD4+ cells (left panel) or CD8+ cells (right panel) stained with isotype antibody (upper panel) or anti-Helios antibodies (lower panel).”
- This discussion poorly strengthens the results of the study. Lines 250-255 are basic textbook data.
Response: We thanks the reviewer’s suggestion and have revised the sentences (on lines 324-328). The sentences before and after revision were shown as the following:
“For instance, CD4+ T naive cells differentiate into T helper 1 (Th1) cells characterized by interferon gamma secretion, which can help CD8+ T cells convert into cytotoxic T cells. However, CD4+ T naive cells differentiate into T helper 2 (Th2) cells (characterized by IL-4, IL-5 and IL-13 secretion), will inhibit Th1 immune responses and hamper antitumor immune responses.”
was changed to
“For instance, CD4+ T naive cells differentiate into T helper 1 (Th1) cells characterized by interferon gamma secretion, which can help CD8+ T cells convert into cytotoxic T cells. However, deficient of Helios increases pTreg differentiation from CD4+ T naive cells, causing the reduction of effector T cells.”
Minor comments:
- Line 27: write “worldwide” at the end of the sentence
Response: The first sentence of the abstract has been revised as the reviewer’s suggestion (on line 27). The sentences before and after revision were shown as the following:
“Immunotherapy is a highly promising approach for the treatment of gastric cancer, the third-leading cause of worldwide overall cancer death.”
was changed to
“Immunotherapy is a highly promising approach for the treatment of gastric cancer, the third-leading cause of overall cancer death worldwide.”
- Lines 33-38: Divide this content into separate sentences
Response: The sentences before and after revision (on lines 33-38) were shown as the following:
“Segregation of gastric cancer patients into high- vs. low-Helios-expressing tumor-infiltrating lymphocytes showed those with high expression to exhibit longer survival in gastric cancer patients, Helicobacter pylori-infected gastric cancer patients and advanced stage (III – IV) gastric cancer patients, especially, Helios expression was an independent factor for survival in advanced gastric cancer patients.”
was changed to
“Segregation of gastric cancer patients into high- vs. low-Helios-expressing tumor-infiltrating lymphocytes showed those with high expression to exhibit longer survival in gastric cancer patients, Helicobacter pylori-infected gastric cancer patients and advanced stage (III – IV) gastric cancer patients. In particular, Helios expression was an independent factor for survival in advanced gastric cancer patients.”
- Line 38: Delete or rephrase: “Except immunohistochemistry”
Response: The sentences before and after revision (on lines 38-40) were shown as the following:
“Except immunohistochemistry, we performed immunofluorescense staining to detect Helios expression in tumor-infiltrating lymphocytes and peripheral blood mononuclear cells.”
was changed to
“We performed immunofluorescense staining to detect Helios expression in tumor-infiltrating lymphocytes and peripheral blood mononuclear cells.”
- Line 59: “It reported” should be ‘it was reported’ or ‘it has been reported’
- Lines 59-69: Reword that sentence
Response: We thanks the reviewer’s suggestion and have reworded the sentences (on lines 59-69). The sentences before and after revision were shown as the following:
“It reported that high levels of TILs associated with positive prognosis, reflective of a protective host antitumor immunity against GC [6]. Additionally, CD3+T cells, including CD8+ and CD4+ T cells, play different roles in tumor immune responses, with activated CD8+T cells becoming cytotoxic T cells, which play a crucial role in determining the clinical outcomes of tumor patients. CD4+ T cells secretes cytokines such as interferon-γ to help CD8+T cells and macrophages, but one particular subset of CD4+T cells- regulatory T (Treg) cells that are essential for immune suppressive function, by inhibiting other effector T cell functions. All these cells interact with others and shape the GC tumor microenvironment.”
was changed to
“It has been reported that high levels of TILs reflected a protective role of host in antitumor immunity against GC and was associated with positive prognosis [6]. Additionally, CD8+ and CD4+ T cells, two subpopulations of CD3+T cells, play different roles in tumor immune responses [7]. The activated CD8+T cells becoming cytotoxic T cells, which play a crucial role in determining the clinical outcomes of tumor patients. CD4+ T cells secretes cytokines such as interferon-γ to activate CD8+T cells and macrophages. Regulatory T (Treg) cells are one particular subset of CD4+T cells that are essential for immune suppressive function, by inhibiting other effector T cell functions. All these cells interact with others and shape the GC tumor microenvironment.”
- Lines 61-64: This is a textbook information, basic knowledge on the field.
Response: We thank the reviewer for this comment. Indeed, we intend to describe basic knowledge of cancer immunology for readers that are not familiar to this field. An additional review paper has also been added as a reference for these information (on line 63).
- Lines 64-67: Divide this content into separate sentences
Response: The sentences before and after revision (on lines 65-68) were shown as the following:
“CD4+ T cells secretes cytokines such as interferon-γ to help CD8+T cells and macrophages, but one particular subset of CD4+T cells- regulatory T (Treg) cells that are essential for immune suppressive function, by inhibiting other effector T cell functions.”
was changed to
“CD4+ T cells secretes cytokines such as interferon-γ to activate CD8+T cells and macrophages. Regulatory T (Treg) cells are one particular subset of CD4+T cells that are essential for immune suppressive function, by inhibiting other effector T cell functions.”
- Myeloid cells, such as MDSCs and TAMs are not mentioned. These cells should be mentioned because these have fundamental effect to maintain the tumor promoting microenvironment (e.g. Inflammation and Cancer: Extra- and Intracellular Determinants of Tumor-Associated Macrophages as Tumor Promoters, Mediators of Inflammation 2017. )
Response: We thanks the reviewer’s suggestion and have revised the sentences (on lines 328-331). The sentences before and after revision were shown as the following:
“Thus, T cell subsets (and their anti-tumor abilities), and combinations with other immune cells, are all important to the constitution of the tumor microenvironment.”
was changed to
“Besides, many suppressor cells such as myeloid-derived suppressor cells, tumor associated macrophages or neutrophils all can play some roles in the constitution of tumor microenvironment to inhibit effective immune responses.”
- Line 71: Change “They” to These, and “perform” to act
Response: The sentences before and after revision (on line 72) were shown as the following:
“They have also been shown to perform as tumor suppressors, because their misexpression in leukemia associates with poor prognosis [9].”
was changed to
“These have also been shown to act as tumor suppressors, because their misexpression in leukemia associates with poor prognosis [10].”
- Change “foxp3” to Foxp3
Response: The “foxp3” has been corrected to “Foxp3” in our manuscript.
- Lines 98-100: This sentence is already in the introduction. This does not fit in the Results section.
Response: We thanks the reviewer’s comments and have deleted two repeated sentences.
- Line 105-106, this sentences is not complete, reword it. ……”malignancies especially a cagA-positove strain”….
Response: We thanks the reviewer’s suggestion and have reworded the sentences (on lines 172-174). The sentences before and after revision were shown as the following:
“Chronic infection with Helicobacter pylori (H. pylori, HP) is an important risk factor for gastric malignancies especially a cagA-positive strain [18].”
was changed to
“Chronic infection with Helicobacter pylori (H. pylori, HP) is an important risk factor especially a cagA-positive strain can enhance the gastric malignancy [21].”
- Reword the Figure Legend 1.
Response: We thanks the reviewer’s suggestion and have reworded the sentences in figure legend 1 (on lines 181-188). The sentences before and after revision were shown as the following:
“Figure 1. Immunohistochemical stains in gastric tumor. Upper panel: Representative immunohistochemical staining of the infiltrated lymphocytes in upper panel was stained with anti-Helios antibody in gastric cancer. (A) and (B) represented low expression of Helios, (C) and (D) represented with high Helios expression. The scale bar in (A) and (C) is 50 μM, and in (B) and (D) is 100 μM. Lower panel: Representative H&E staining of the infiltrated lymphocytes in gastric cancer. (E) and (F) represented low expression of Helios, (G) and (H) represented with high Helios expression. The scale bar in (E) and (G) is 50 μM, and in (F) and (H) is 100 μM.”
was changed to
“Figure 1. Immunohistochemical and H&E stains in gastric tumor. (A-D) represented immunohistochemical staining using anti-Helios antibody in gastric tumor tissue. Low expression of Helios in gastric cancer was represented by (A) (100X) and (B) (400X). High expression of Helios in gastric cancer was represented by (C) (100X) and (D) (400X). (E-H) represented H&E staining in gastric tumor tissue. Low expression of Helios in gastric cancer was represented by (E) (100X) and (F) (400X). High expression of Helios in gastric cancer was represented by (G) (100X) and (H) (400X). The scale bar in (A) and (C) were 50 μM, and others were 10 μM.”
- Line 162: Check that paradox: „It did not 162 show significant differences. The test resulted in a P value <0.05”. The p value under 0.05 generally means one star significance.
Response: We thank the reviewer’s mention in the error, and the sentence “It did not show significant differences.” has been moved into the end of the legend of Figure 3B (on line 230). The sentences before and after revision were shown as the following:
“It did not show significant differences. The test resulted in a P value <0.05. (B) Kaplan–Meier curves of overall survival of lower stage (T1+T2) gastric cancer patients obtained by the expression of Helios in tumor-infiltrating lymphocytes in tumor tissues.”
was changed to
“The test resulted in a P value <0.05. (B) Kaplan–Meier curves of overall survival of lower stage (T1+T2) gastric cancer patients obtained by the expression of Helios in tumor-infiltrating lymphocytes in tumor tissues. It did not show statistic differences.”
- Line 204: Write human instead of “man”.
Response: The sentences before and after revision (on lines 278-279) were shown as the following:
“In Treg cells, Helios is expressed in 60–70% of Treg cells in both mouse and man [22],”
was changed to
“Helios is expressed in 60–70% of Treg cells in both mouse and human [25].”

Round 2
Reviewer 2 Report
The manuscript has been improved and can be accepted. Minor grammaticar errors should be checked in the revised version.
Author Response
Response: We thank the reviewer’s comment and have revised the sentences as shown below.
On the line 63:
“The activated CD8+T cells becoming cytotoxic T cells,”
was changed to
“The activated CD8+T cells become cytotoxic T cells,”
On the lines 174-176:
“In a subgroup of H. pylori-positive infected GC patients, the high TIL Helios expression group showed significantly longer survival than those in the low Helios expression group (Figure 2B, log rank test, P=0.02).”
was changed to
“In a subgroup of H. pylori-positive infected GC patients, the high Helios-expressing TIL group showed significantly longer survival than those in the low Helios-expressing TIL group (Figure 2B, log rank test, P=0.02).”
On the line 276:
“Ikaros, the founding member of this family, mainly expressed in most hematopoietic cells, whereas Helios is expressed primarily in T-lineage cells and early multipotential precursor cells”
was changed to
“Ikaros, the founding member of this family, is mainly expressed in most hematopoietic cells, whereas Helios is expressed primarily in T-lineage cells and early multipotential precursor cells”
